# Assessment of Treatment Response after Pressurized Intra-Peritoneal Aerosol Chemotherapy (PIPAC) for Appendiceal Peritoneal Metastases

**DOI:** 10.3390/cancers14204998

**Published:** 2022-10-12

**Authors:** SP Somashekhar, Julio Abba, Olivia Sgarbura, Mohammad Alyami, Hugo Teixeira Farinha, Ramya G. Rao, Wouter Willaert, Martin Hübner

**Affiliations:** 1Manipal Comprehensive Cancer Center, Manipal Hospital, HAL Old Airport Rd, Kodihalli, Bengaluru 560017, India; 2Department of Digestive and Emergency Surgery, Grenoble Alpes University Hospital, CEDEX 09, F-38043 Grenoble, France; 3Surgical Oncology Department, Montpellier Cancer Institute (ICM), University of Montpellier, F-34298 Montpellier, France; 4Institut de Recherche en Cancérologie de Montpellier (IRCM), INSERM U1194, Université de Montpellier, F-34298 Montpellier, France; 5Department of General Surgery and Surgical Oncology, Oncology Center, King Khalid Hospital, Najran 66262, Saudi Arabia; 6Department of Visceral Surgery, Faculty of Biology and Medicine UNIL, Lausanne University Hospital (CHUV), Rue du Bugnon 46, 1011 Lausanne, Switzerland; 7Department of GI Surgery, Ghent University Hospital, 9000 Ghent, Belgium

**Keywords:** peritoneal regression grading system, PRGS, PIPAC, peritoneal metastasis, chemotherapy, survival, RECIST

## Abstract

**Simple Summary:**

Pressurized IntraPeritoneal Aerosol Chemotherapy (PIPAC) is an emerging treatment modality for patients with peritoneal cancer with good safety profile and promising early response rates. The aim of this study was to analyze survival and surrogates for oncological response after PIPAC ± systemic chemotherapy for appendiceal tumours. Median overall survival of this cohort was 30 months from time of diagnosis and 22 months from PIPAC1 (per protocol) comparing favorably with 20.4 months of OS reported for patients with palliative chemotherapy alone. However, without prospective comparative data, the role of PIPAC for appendicular cancer with peritoneal metastases remains unclear.

**Abstract:**

Background The aim of this study was to analyse survival and surrogates for oncological response after PIPAC for appendiceal tumours. Methods This retrospective cohort study included consecutive patients with appendiceal peritoneal metastases (PM) treated in experienced PIPAC centers. Primary outcome measure was overall survival (OS) from the date of diagnosis of PM and from the start of PIPAC. Predefined secondary outcome included radiological response (RECIST criteria), repeat laparoscopy and peritoneal cancer index (PCI), histological response assessed by the Peritoneal regression grading system (PRGS) and clinical response. Results Final analysis included 77 consecutive patients (208 PIPAC procedures) from 15 centres. Median OS was 30 months (23.00–46.00) from time of diagnosis and 19 months (13.00–28.00) from start of PIPAC. 35/77 patients (45%) had ≥3 procedures (pp: per protocol). Objective response at PIPAC3 was as follows: RECIST: complete response 4 (11.4%), 11 (31.4%) partial/stable; mean PRGS at PIPAC3: 1.8 ± 0.9. Median PCI: 21 (IQR 18–27) vs. 22 (IQR 17–28) at baseline (*p* = 0.59); 21 (60%) and 18 (51%) patients were symptomatic at baseline and PIPAC3, respectively (*p* = 0.873). Median OS in the pp cohort was 22.00 months (19.00–NA) from 1st PIPAC. Conclusion Patients with PM of appendiceal origin had objective treatment response after PIPAC and encouraging survival curves call for further prospective evaluation.

## 1. Introduction

Peritoneal metastases (PM) from appendiceal tumours are a distinct entity of peritoneal disease with important differences in terms of prognosis and treatments compared to colorectal tumours [1,2]. The estimated incidence of cancers and tumors (neoplasms) of the appendix is 0.15–0.9 per 100,000 people [3]. The literature on appendiceal neoplasms consists due to its rarity mostly of retrospective studies with limited sample size and risk for selection bias. Metastatic appendiceal cancer has a bad prognosis with 5-year OS between 18% to 19% under 5-FU-based palliative chemotherapy with or without combination of capecitabine and oxaliplatin [1,4,5].

Complete cytoreductive surgery (CRS) ± Hyperthermic Intraperitoneal Chemotherapy (HIPEC) is the preferred treatment option for resectable patients offering favourable survival in selected patients [6,7]. Systemic chemotherapy can be added in analogy with colorectal adenocarcinoma although the molecular profile of appendiceal adenocarcinoma is different from colorectal with implications for response to systemic treatment [8,9]. Therapeutic options are limited for patients unfit for major surgery, or those with relapse refractory to systemic treatment options. Retrospective reports of OS after palliative treatment alone with systemic chemotherapy (in majority of cases with capecitabine or fluorouracil) showed a median OS up to 20.4 months [10,11]. After CRS-HIPEC for mucinous appendiceal primaries, 5-year OS for the low- and high-grade mucinous cohorts was 62.5% and 37.7%, respectively. A 5-year survival for the high-grade group who had a complete cytoreduction was of 45% for patients with PCI > 20 and 66% for patients with PCI < 20. High-grade non mucinous appendiceal primaries including adenocarcinoma, goblet cell, and carcinoid tumors derive significantly less benefit from a CRS-HIPEC procedure, with a 3-year survival of approximately 15% [1,5,12,13].

Pressurized IntraPeritoneal Aerosol Chemotherapy (PIPAC) was suggested as an alternative in the palliative situation combining minimal-invasive approach, enhanced pharmacokinetic properties and repeated administration of IP chemotherapy [14,15]. The existing evidence suggests a favourable safety profile and promising reponse rates but these results come mainly from single-center experiences on different tumour entities [16,17,18].

The aim of this multicenter study was to study survival and various surrogates for treatment response after PIPAC specifically for PM of appendiceal origin.

## 2. Materials and Methods

This is a multicenter retrospective cohort study on consecutive PIPAC patients treated for PM of appendiceal origin. All PIPAC centres having performed more than 60 procedures in total by November 2018 were contacted for participation and no center was deliberately excluded [19]. Exclusion criteria were other tumor entities, patient refusal, and patients treated outside current indications. Low-grade appendiceal mucinous neoplasms (LAMN) and High-grade appendiceal mucinous neoplasms (HAMN) were not included into this analysis [16]. The study was conducted according to the declaration of Helsinki (IRB approval: #ICM-ART-2020/05).

### 2.1. Pressurized IntraPeritoneal Aerosol Chemotherapy

Surgical technique, safety and treatment protocols underlie little variation between centers according to recent investigations [18,19] due to a standardized training curriculum [20]. Main features of PIPAC treatment including technical aspects have been summarized recently [21] and include a two-trocar technique (balloon trocars), a standardized safety procotol (advanced ventilation system, zero flow, remote application), and constant pressure conditions of 12 mmHg [22,23]. The diagnostic phase includes documentation of disease extent (Peritoneal cancer index: PCI), aspiration of ascites (volume, cytology) and 3–4 biopsies from different areas of the abdomen for grading of histological response (outlines below). The empirical drug regimen used for most patients was oxaliplatin at 92 mg/m^2^.

### 2.2. Outcomes Measures

Primary endpoint was OS from first PIPAC and from diagnosis of PM. Predefined subgroup analysis was performed for patients having received at least 3 PIPAC treatments (per protocol; pp). Secondary outcome measures were all available potential surrogates for treatment response: symptoms, quality of life (QoL), repeated documentation of PCI [24], radiological response according to Response Evaluation Criteria in Solid Tumours (RECIST) [25] and histological response by use of the peritoneal regression grading score (PRGS) [26,27]. Symptoms were accounted as dichotomous variables including abdominal pain, distension, nausea, and altered intestinal transit (including obstruction) [16,18]. QoL was assessed by use of the validated EORTC QLQ-C30 survey analysing overall QoL as well as its components and main symptoms [28]. Repeated imaging was performed before, during (mostly after 2nd PIPAC) and after treatment, mainly by computed tomography or by magnetic resonance imaging (MRI) and positron emission tomography (PET)/CT if indicated. Treatment response was assessed by use of RECIST criteria [25]. Assessment of histological response and cytology were performed during repeated PIPAC procedures. Aspiration of ascites or peritoneal washing to be sent for cytology and conversion of positive (presence of malignant cells) to negative cytology was counted as treatment response. In addition, 3–4 representative biopsies of PM were performed during PIPAC procedures and analysed according to PRGS. PRGS was strongly propagated and encouraged after its proposal in 2016 [26] and validated recently [27]. PRGS evaluates the histological response of treatment on PM by evaluating the number of tumor cells, fibrosis, acellular mucin pools and necrosis. As described by Solass et al., the PRGS score is defined as follows: 1 corresponds to a complete regression with absence of tumor cells; 2 to major regression features with only a few residual tumor cells; 3 to minor regression with predominance of residual tumor cells and only few regressive features; and 4 corresponds to an absence of response to therapy and where the tumor cells are not accompanied by any regressive features. A PRGS was assessed for each biopsy taken during each PIPAC procedure. The mean PRGS (out of a minimum of 4 biopsies) is calculated according to current recommendations in order to illustrate overall histological response [26,27].

### 2.3. Statistics

For the descriptive analysis, Student’s t-test for continuous data, Kruskal-Wallis test for non-continuous data and a chi-squared test for categorical data was performed. Descriptive statistics are expressed as mean ± SD, median (IQR) or n (%). Repeated measures *t*-test was performed for comparing means before and after treatment. Overall survival from the time of diagnosis and from first PIPAC were calculated by use of the Kaplan–Meier method. Variables for survival outcomes were fitted to univariate Cox models, and multivariate Cox models were then created using forward selection strategy. The assumption of proportional hazard was tested. The statistical significance level was considered as <0.05. For all the statistical analysis Statistical software RStudio (Version 1.4.1106) was used. Percentages were calculated based on the availability of information and not to the total number of patients per group.

## 3. Results

A total of 77 patients from 15 centres having 208 PIPAC procedures were included in the analysis. 35 patients had ≥3 PIPAC treatments per protocol (pp) as detailed in the patient flow chart (Figure 1). Patients’ and tumour characteristics overall and for the pp cohort are displayed in Table 1. PIPAC was applied as monotherapy in 50 (65%) patients, while it was combined with systemic therapy in 27 (35%) patients.

Median follow-up from diagnosis and from start of PIPAC treatment was 20.6 (IQR 13.5–32.8) months and 9.8 (IQR 3.9–21.1) months, respectively. Figure 2A,B depict OS of patients with PM of appendiceal origin computed from the date of diagnosis and from PIPAC1, respectively.

For the pp cohort, OS from diagnosis and start of PIPAC treatment were 33 months (19.00–NA) and 22 months (19.00–NA) respectively (Figure 2C,D). Figure 3 depict OS of patients computed from PIPAC1 stratified by PCI < 11, 11–20, >20.

Variables indicating treatment response are displayed for both cohorts in Table 2. No significant difference was observed in pp cohort for parameters such as cytology, ΔPCI and presence of any symptoms when compared to the values at base line.

In univariate analysis, total number of cycles of chemotherapy, HR 0.75 (95% CI 0.6–0.93), CEA pre PIPAC, HR1 (95% CI 1–1), Albumin pre –PIPAC, HR 0.8 (95% CI 0.66–0.96), ascites at first PIPAC, HR 1 (95% CI 1–1) and patients who showed radiological response after 3rd PIPAC, HR 3.7(95% CI 1–14) were significant predictors for overall survival (Appendix A). However, none of these factors was retained after multivariate analysis (Appendix A).

## 4. Discussion

This international cohort of patients with appendiceal cancer peritoneal metastases showed encouraging survival results and objective response after repeated PIPAC treatment.

The estimated incidence of tumors of the appendix is 0.15–0.9 per 100,000 people [29]. Therefore, most of the existing reports in [16] the literature are small retrospective studies with high risk for potential selection bias. Systemic chemotherapy and CRS ± HIPEC are part of current management protocols [12,30], but the sequence of recommend treatments is inconsistent [6].

The histopathological classification of appendiceal tumors evolved during the last two decades [31]. The most recent classifications return to a three-scale grading system for all appendiceal neoplasms where G1 is a low-grade appendiceal neoplasms that features a particularly favourable prognostic. However, given the moment of the data collection for this study that preceded the WHO 2019 classification, our grading refers to the mucinous adenocarcinoma alone [32]. The patients with adenocarcinoma have significantly worse prognostic compared to the other entities [2].

This multicentric study includes patients from diverse geographic areas of the world with inherent differences with regards to the socio-economic situation and healthcare infrastructure. Common features of the participating centers are special expertise in peritoneal surface malignancies and similar indications and treatment protocols, reserving PIPAC for patients with unresectable disease, mostly beyond the first line.

Currently there is no standard for systemic chemotherapy regimen for surgically unresectable patients diagnosed with metastatic appendiceal cancer. Many chemotherapy regimens have been extrapolated from the metastatic colorectal population and are largely 5-fluorouracil based [32]. However, there are important molecular differences described between the two entities that explain the lower rates of response and survival in the appendiceal cancer population [2]. The latter also has higher rates of signet ring cells (SRC) subtype that are associated with ominous prognosis in any primary [33]. In spite of the absence of specific chemotherapy protocols, it has been shown that patients with mucinous adenocarcinoma (MAC), non-mucinous adenocarcinoma (NMAC) and SRC all benefit from adjuvant chemotherapy, while there is little data about the results of induction treatment [34,35].

In this series, 80% of the patients received oxaliplatin-based treatment in their 1st chemotherapy line with irinotecan-based treatment in the 2nd line. In a retrospective analysis by Shapiro et al. systemic chemotherapy prolonged disease control by 7.6 months and OS up to 20.4 months in patients who are deemed suboptimal candidates for CRS +/− HIPEC [10,11,12,30,31,36]. In comparison to this, the median OS in our study was 20.9 months (13.7–31.4) from diagnosis and 9.9 months (4.5–20.8) from the time of the first PIPAC. For the pp cohort, the median OS was 22 months (19.00-NA) from first PIPAC. The current results seem encouraging, and they suggest that the use of PIPAC in this setting can further be explored as stand-alone treatment or in combination with systemic chemotherapy (bi-directional). The latter poses obvious methodological challenge, namely to attribute potential benefits of the combined treatment to either modality, as it was the case also in the present study. Further evidence arising from large registry data is expected in the future.

IP chemotherapy allows a higher drug concentration intraperitoneally compared to systemic chemotherapy, resulting In better response in terms of peritoneal metastasis, along with less systemic toxicity [37]. The pp cohort set of patients in this study showed an improved survival benefit when compared to patients who received less than 3 cycles of PIPAC: median OS from time of first PIPAC of 22 months (19.00-NA) vs 10 months (8.00-NA) (*p*-value = 0.13). However, the results did not reach statistical significance (potential type II error). The survival benefit was even higher in patients who prior received two lines of chemotherapy as it gained statistical significance (28 months versus 8 months, *p*-value = 0.02).

The potential role of neoadjuvant intraperitoneal chemotherapy was already emphasized in other appendiceal entities (pseudomyxoma) [38]. The probable effectiveness of other types of intraperitoneal administrations in the context of peritoneal disease of appendiceal origin adds to the rationale of PIPAC, in spite of the clinical setting of the present study that is limited to appendiceal adenocarcinoma [12,33].

In the current study we mostly used oxaliplatin as a PIPAC drug. This attitude is still considered up-to-date by the recent consensus on drug regimens [39]. On the other side, the recent literature questions the role of oxaliplatin as an IP drug after oxaliplatin-based neoadjuvant treatment due to potential resistance. The current data concerning the potential resistance are scarce and have low-quality, and PIPAC can potentially overcome some of the limitations of HIPEC through repeated administration of low-dose, long duration IP treatment [40]. Furthermore, in the particular setting of resectable appendiceal adenocarcinoma, a randomized control trial showed similar survival results with HIPEC with oxaliplatin and mitomycine C but HIPEC with oxaliplatin was associated with a better quality of life [30,41]. All these aspects indirectly support the further use of PIPAC-Ox in the metastatic appendiceal adenocarcinoma patients while pursuing efforts to identify new IP drugs.

Although on multivariate regression analysis we could not identify any significant variables that can predict survival in the pp cohort, factors that showed promising trends were radiological response at PIPAC 3, tumour markers CA 19-9 and CA 125. PRGS was expected to confirm its prognostic nature for PM similar to histologic response in other metastatic sites [42]. In this cohort, it failed to show a significant signal which suggest that a more complex use of the score may be required in order to enhance its value. Small sample size and missing data inherent to the retrospective and multicentric nature of the study are probable causes for the lack of more conclusive results.

The main limitations are the retrospective study design, the small and heterogeneous cohort of patients and the limited follow-up time. However, the patient demographics, disease presentation, and treatments are consistent with the literature and can hence be considered as representative. Despite the limitations, the current data is the best available evidence on this topic and it can guide the management of the disease in the adapted clinical context. However further prospective studies are needed in order to further elucidate the role of PIPAC for appendiceal PM.

## 5. Conclusions

In conclusion, PIPAC appears to be a promising treatment option for patients with PM of appendiceal origin. PIPAC can hence be discussed for patients in this situation with no standard treatment option available. Large-scale registry data and prospective comparative data are needed to confirm oncological efficacy before use of PIPAC can be validated for this indication and potential others.

## Figures and Tables

**Figure 1 cancers-14-04998-f001:**
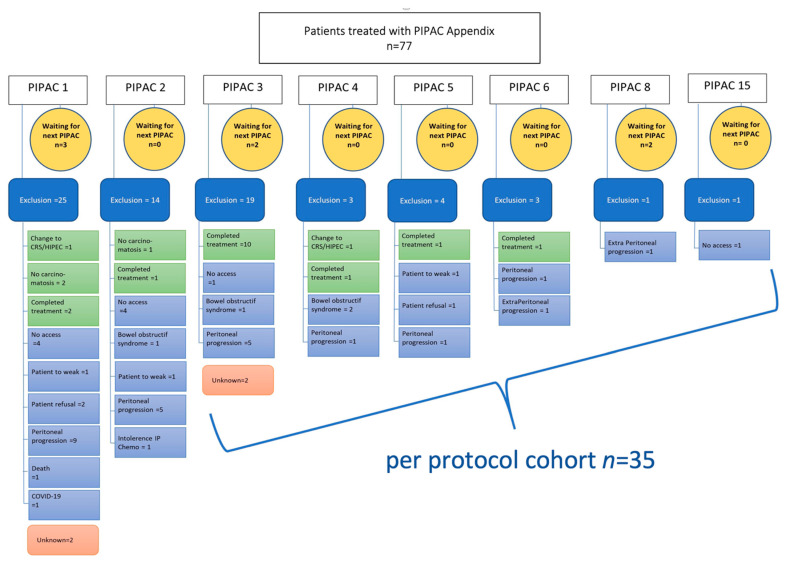
Flow chart of all patients undergoing PIPAC procedures. Causes of PIPAC treatment interruption are described precisely. PIPAC = Pressurized IntraPeritoneal Aerosol Chemotherapy; CRS = Cyto Reductive Surgery, HIPEC = Hyperthermic intraperitoneal chemotherapy; IP = intraperitoneal chemotherapy.

**Figure 2 cancers-14-04998-f002:**
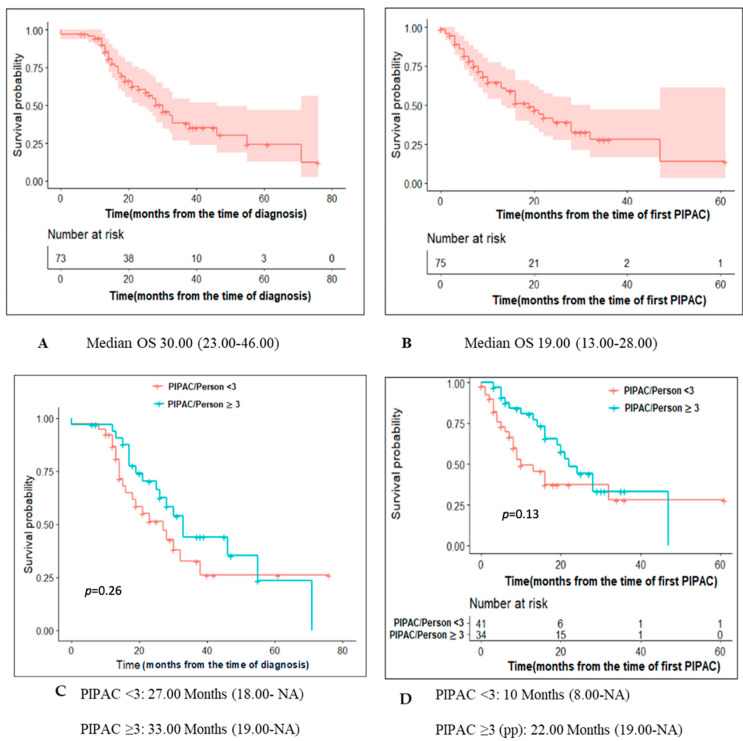
Survival of patients undergoing Pressurized IntraPeritoneal Aerosol Chemotherapy for peritoneal metastases of appendiceal origin. OS for the entire cohort from time of diagnosis (**A**) and first PIPAC (**B**). By protocol analysis for OS (**C**) from diagnosis and OS (**D**) from first PIPAC.

**Figure 3 cancers-14-04998-f003:**
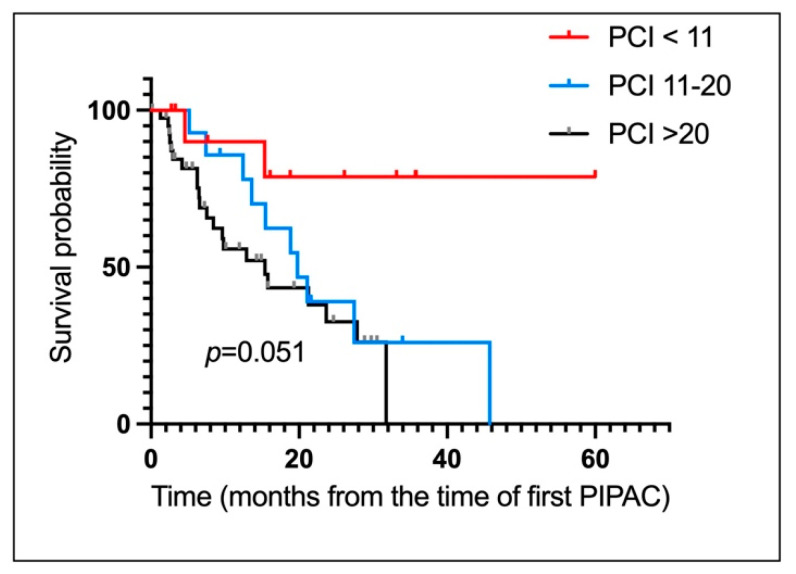
Survival of patients undergoing Pressurized IntraPeritoneal Aerosol Chemotherapy for peritoneal metastases of appendiceal origin. OS for the entire cohort from time first PIPAC stratified by PCI. (PCI < 11, 11–20, >20). PCI < 11: NA (5.00-NA). PCI 11–20: 20.00 Months (14.00–29.00). PCI: >20: 15.00 Months (9.00–22.00).

**Table 1 cancers-14-04998-t001:** Baseline characteristics of patients undergoing Pressurized IntraPeritoneal Aerosol Chemotherapy for appendiceal peritoneal metastases.

Parameter		All Patients (*n* = 77)	PP Cohort		*p* Value
			<3 PIPACs (*n* = 42)	≥3 PIPACs (*n* = 35)	
Median Age (IQR)		56.7 (47.1–66.2)	56.8 (47.7–65.4)	56.8 (47.0–66.6)	0.999
Age group, *n* (%)	≤30	2 (2.56)	2 (4.76)	0 (0)	0.545
	31–40	4 (5.13)	1 (2.38)	3 (8.57)	
	41–50	19 (24.36)	10 (23.81)	9 (25.71)	
	51–60	25 (32.05)	15 (35.71)	10 (28.57)	
	61–70	15 (19.23)	7 (16.67)	8 (22.86)	
	>70	13 (16.67)	8 (19.05)	5 (14.29)	
Gender, *n* (%)	Male	37 (48.72)	24 (57.14)	13 (37.14)	0.080
	Female	40 (51.28)	18 (42.86)	22 (62.86)	
Median BMI (kg/m^2^) (IQR)		22.86 (20.32–25.66)	23.15 (20.33–25.70)	22.93 (20.78–25.39)	0.934
ASA	1	8 (12.9)	4 (9.52)	4 (11.43)	0.149
	2	30 (48.39)	12 (28.57)	18 (51.43)	
	3	24 (38.71)	16 (38.10)	8 (22.86)	
ECOG	0	28 (43.08)	15 (35.71)	13 (37.14)	0.749
	1	25 (38.46)	14 (33.33)	11 (31.43)	
	2 + 3	12 (18.42)	6 (14.20)	6 (17.14)	
Pathology	Synchronous	64 (85%)	32 (80%)	32 (91%)	0.163
	Metachronous	11 (15%)	8 (20%)	3 (91%)	
Histology	G1	18 (33%)	9 (32%)	9 (33%)	0.856
	G2	14 (25%)	8 (29%)	6 (22%)	
	G3	23 (42%)	11 (39%)	12 (44%)	
RAS	No	10 (33%)	6 (43%)	4 (25%)	0.301
	Yes	20 (67%)	8 (57%)	12 (75%)	
Previous CRS + HIPEC	No	63 (82%)	31 (72%)	32 (94%)	0.013
	Yes	14 (18%)	12 (28%)	2 (6%)	
Previous CRS	No	50 (65%)	26 (60%)	24 (71%)	0.355
	Yes	27 (35%)	17 (40%)	10 (29%)	
Previous 1st chemo cycle	No	9 (12%)	5 (12%)	4 (12%)	0.985
	Yes	68 (88%)	38 (88%)	30 (88%)	
	Oxaliplatin based	51 (80%)	29 (85%)	22 (73%)	0.663
	Irinotecan based	6 (9%)	2 (6%)	4 (13%)	
	Oxiri based	5 (8%)	2 (6%)	3 (10%)	
	Biological therapy	24 (38%)	11 (32%)	13 (43%)	0.365
	Total cycle (IQR)	8 (6–12)	8 (6–12)	8 (6–12)	0.818
Previous 2th chemo cycle		32 (45%)	15 (38%)	17 (53%)	0.217
Previous 3th chemo cycle		9 (13%)	5 (13%)	4 (13%)	0.983
Total cycles (IQR)		11 (6–14)	10 (6–14)	12 (6–14)	0.772
Bimodal (PIPAC + IV chemo)		27 (35%)	10 (24%)	17 (49%)	0.042
Median PCI at Baseline		23 (14–30)	23 (12–31)	22 (16–28)	0.948
Total cycles	≤12	38 (76%)	20 (71%)	18 (82%)	0.393
	>12	12 (24%)	8 (29%)	4 (18%)	
Symptoms prePIPAC	No	33 (43%)	19 (45%)	14 (40%)	0.644
	Yes	44 (57%)	23 (55%)	21 (60%)	
Pain	No	43 (61%)	24 (65%)	19 (56%)	0.439
	Yes	28 (39%)	13 (35%)	15 (44%)	
Ascites	No	54 (70%)	27 (64%)	27 (77%)	0.220
	Yes	23 (30%)	15 (36%)	8 (23%)	
Dysphagia	No	67 (97%)	34 (94%)	33 (100%)	0.346
	Yes	2 (3%)	2 (6%)	0 (0%)	
Obstructive symptoms	No	65 (92%)	34 (92%)	31 (91%)	0.914
	Yes	6 (8%)	3 (8%)	3 (9%)	
Nausea	No	60 (85%)	30 (81%)	30 (88%)	0.405
	Yes	11 (15%)	7 (19%)	4 (12%)	
CEA (µg/l) (SD)		26.4 ± 52.7	26.0 ± 53.6	26.7 ± 52.9	0.953
Ca19.9 (U/mL) (SD)		291.4 ± 645.9	449.9 ± 823.6	101.2 ± 242.6	0.020
Ca125 (U/mL) (SD)		109.3 ± 129.2	190.2 ± 134.1	28.3 ± 51.6	0.001
Creatinin (µmol/L) (SD)		73.2 ± 22.6	71.6 ± 18.5	74.9 ± 26.7	0.524
Albumin (g/L) (SD)		39.4 ± 9.7	39.9 ± 13.3	39 ± 4.4	0.706

Median (IQR—Interquartile Rang or Range), Mean (SD—Standard Deviation) or number (%) as appropriate. Statistical significance (*p* < 0 05) is highlighted in bold. PP cohort = per protocol cohort, ECOG = Eastern Cooperative Oncology Group, CRS = Cyto Reductive Surgery, HIPEC = Hyperthermic intraperitoneal chemotherapy, CEA = Carcinoembryonic antigen, SD = standard deviation; BMI = body mass index; IQR = interquartile range; PIPAC = Pressurized IntraPeritoneal Aerosol Chemotherapy.

**Table 2 cancers-14-04998-t002:** Treatment response of patients undergoing Pressurized IntraPeritoneal Aerosol Chemotherapy for peritoneal metastases of appendiceal origin.

Parameter		PP Cohort *n* = 35		*p* Value
		at Baseline	≥3 PIPACs	
RECIST	Regression/Stable	-	15 (43%)	-
	Progression	-	5 (25%)	
PRGS	1–2	-	17 (49%)	-
	3–4	-	5 (25%)	
Cytology	Positive	7 (20%)	3 (9%)	0.606
	Negative	28 (80%)	32 (91%)	
PCI		24 (18–29)	21 (18–28)	0.104
ΔPCI (PIPAC1 vs. 3)	≥3 decrease	-	13 (37%)	0.113
	<3 or increase	-	18 (51%)	
Any Symptoms	Yes	21 (60%)	18 (51%)	0.873
	No	14 (40%)	17 (49%)	

Median (IQR—Interquartile Rang or Range), Mean (SD—Standard Deviation) or number (%) as appropriate. Statistical significance (*p* < 0 05) is highlighted in bold. PCI: Peritoneal Cancer Index.

## Data Availability

Not applicable.

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
