# Peer review of "Assessment of Treatment Response after Pressurized Intra-Peritoneal Aerosol Chemotherapy (PIPAC) for Appendiceal Peritoneal Metastases"

_cancers, 2022, doi:10.3390/cancers14204998_

Round 1

Reviewer 1 Report

This manuscript present the results of an multi-centric analysis of treatment response following the administration of pressurized intra-peritoneal aerosol chemotherapy (PIPAC) in patients with peritoneal metastasis from appendiceal cancer. This can be used as an adjuvant method to systemic chemotherapy or as a sole treatment option. The text is well structured and easy to follow. However, there are a few limitations to this study, particularly due to the small sample size and the consequent heterogeneity, as the authors mention. Due to this aspect, I would recommend a more cautious formulation of the conclusions, as they can sound a bit overoptimistic. I would also suggest to the authors to include in the results/discussion section a phrase on post PIPAC related complications, as these can sometimes preclude further treatment. 

Minor observations: -Define abbreviations at first use in the abstract; - The title for Figure 1 should be more explanatory.

Author Response

This manuscript presents the results of a multi-centric analysis of treatment response following the administration of pressurized intra-peritoneal aerosol chemotherapy (PIPAC) in patients with peritoneal metastasis from appendiceal cancer. This can be used as an adjuvant method to systemic chemotherapy or as a sole treatment option. The text is well structured and easy to follow. However, there are a few limitations to this study, particularly due to the small sample size and the consequent heterogeneity, as the authors mention. Due to this aspect, I would recommend a more cautious formulation of the conclusions, as they can sound a bit overoptimistic. I would also suggest to the authors to include in the results/discussion section a phrase on post PIPAC related complications, as these can sometimes preclude further treatment. 

Minor observations: -Define abbreviations at first use in the abstract; - The title for Figure 1 should be more explanatory.

Many thanks to the reviewer for the time and efforts invested!

Appendiceal tumors are a rare and heterogeneous tumor, and evidence is hence limited for all kind of treatments unfortunately. This is true in particular for new approaches such as PIPAC. So far, no specific evidence was available for PIPAC of appendiceal tumors. This report is therefore a small step forward despite the limitations: small study sample, heterogeneity. This was emphasized in the discussion and the conclusions were phrased with more caution as requested.

PIPAC related complications are rare, and this was repeatedly and consistently reported in the literature (refs). Main reasons for stopping PIPAC are related to disease progression as outlined in the study flow chart and reported by others. Hereford, improved patient selection is of utmost importance and a prediction score is currently elaborated and validated by the international collaborative group.

Reviewer 2 Report

Dear authors -

Thank you for this updated outcomes paper on PIPAC. Comments below for revisions:

A.     Quality of scientific content

a.      General criteria

                                                    i.     Keywords: non included

b.      Introduction/Background

                                                    i.     Please review survival for peritoneal disease using known treatment lines:

1.      CRS alone

2.      Chemotherapy alone

3.      CRS with chemotherapy

4.      Any other relevant modalities with published information

5.      Any prior data using PIPAC for peritoneal malignancies

6.      All survival data from previous treatment modalities should be reported

                                                   ii.     What does this trial add to the body of knowledge on PIPAC?

c.      Methods

                                                    i.     What is the difference in mucinous and non-mucinous types of appendiceal carcinoma? I do not see this as a variable.

                                                   ii.     What was PCI prior to PIPAC?

1.      Can you segment patients by pre-therapy PCI score in the OS analysis?

                                                  iii.     There are two common and important variables, further may follow upon revisions

d.      Results

                                                    i.     Ok, will need to be re-assessed after changes above

e.      Discussion

                                                    i.     Discuss results with survival with respect to other modes of treatment

                                                   ii.     Comment on results compared to Bromeline/NAC treatment

1.      Can this be utilized with PIPAC?

f.       Conclusion

                                                    i.     Ok

B.     Originality and innovativeness

a.      Updates to trials

C.      Coverage of related literature

a.      References

                                                    i.     Requires improvement

D.     Organization and clarity

a.      Abstract

                                                    i.     Rewrite based on changes to methods and analysis

b.      Paper

                                                    i.     Please see above comments

E.      Conflict of interest and ethical issues

a.      Conflicts of interest

                                                    i.     Include this statement

b.      Human subject research

                                                    i.     ok

Looking forward to your paper upon revisions.

Author Response

Thank you for this updated outcome paper on PIPAC. Comments below for revisions:

Thanks very much to the reviewer for a very careful revision and insightful comments.

Quality of scientific content

General criteria: Keywords: non included – Keywords were added

Introduction/Background

Please review survival for peritoneal disease using known treatment lines; CRS alone; Chemotherapy alone; CRS with chemotherapy; Any other relevant modalities with published information; Any prior data using PIPAC for peritoneal malignancies; All survival data from previous treatment modalities should be reported

Direct evidence is very limited for appendicular tumors and most recommendations are based on indirect evidence coming from trials in colorectal cancer. Key information and refs were added to the introduction.

What does this trial add to the body of knowledge on PIPAC?

This is the first report on PIPAC treatment for appendicular tumors with important information on treatment response and survival. More data is needed for sure but RCTs are unlikely to be performed for this heterogeneous group of tumors.

Methods

What is the difference in mucinous and non-mucinous types of appendiceal carcinoma? I do not see this as a variable.

This variable was unfortunately not available in many of the institutional databases and could hence not be reported.

What was PCI prior to PIPAC?

PCI at baseline was added to the manuscript.

Can you segment patients by pre-therapy PCI score in the OS analysis?

There are two common and important variables, further may follow upon revisions – PCI has been shown to be an important predictor for OS in the CRS/HIPEC literature. Post hoc subgroup analyses are problematic, and we would propose not to include this information in the paper. Nonetheless, we prepared upon request a survival analysis stratified by PCI <11, 11-20, >20.

Results

Ok, will need to be re-assessed after changes above

Discussion

Discuss results with survival with respect to other modes of treatment

PIPAC and CRS/HIPEC are not used for the same indications and comparisons of survival between curative and palliative approaches are unusual. The evidence of palliative chemotherapy for appendicular tumors is also very limited and most treatment recommendations derive from indirect evidence from colorectal cohorts. The best available evidence was added to the discussion and introduction as requested by the reviewer to give an overview on different modalities to the reader.

Comment on results compared to Bromeline/NAC treatment

Bromaline/NAC is a promising experimental approach to treat inoperable or non-resectable patients with pseudomyxoma peritonei. This is a very different situation and concerns a different disease entity.

Round 2

Reviewer 2 Report

Dear authors - 

Thank you for your hard work on this interesting retrospective review of results. Although there have been improvements, there are still elements that require clarification as described in the attachment.

1. Use a standard measure of survival e.g. median OS months or 5 year OS % in introduction / results

2. Include data on palliative treatment alone e.g. chemotherapy

3. What is post-treatment PCI (this was collected per your protocol)?

4. PFS?

5. Multivariate Cox Proportional Hazard analysis or similar given heterogeneity in study groups e.g. (CRS/HIPEC, bimodal IV chemo, etc.)

6. For similar pathology and PCI score with differing treatments was is the objective discussion of your results?

7. PMP is usually of appendiceal carcinoma of mucinous origin (in fact you reference this in your paper). Include a comment on bromlein/NAC.

Thank you,

Author Response

We would like to acknowledge the detailed reviews of the referees which helped us to improve our manuscript. We understand that reviewer 1 had no further comments. The following provides our reply to remaining issues raised by reviewer 2:

1. Use a standard measure of survival e.g. median OS months or 5 year OS % in introduction / results

According to the reviewer’s request we added OS for CRS/HIPEC in the introduction, which was mostly reported as 3 or 5y OS (%). As mostly done for palliative treatments with limited survival, we presented the data of our study as median survival (months)

2. Include data on palliative treatment alone e.g. chemotherapy

The requested information was added to the introduction and discussion giving median OS for palliative chemotherapy as benchmark.

3. What is post-treatment PCI (this was collected per your protocol)?

PCI remains controversial as potential marker for treatment response. Nonetheless, we added this information in addition to delta PCI (already given) to Table 2.

4. PFS?

The definition and assessment of PFS for peritoneal metastases of appendicular origin and peritoneal disease in general remains very controversial. The authors agree with other experts, that PFS is not an ideal endpoint for these studies. Moreover, and probably for this very reason, PFS was not assessed in the prospective databases of most of the participating centers and was hence not available here for analysis.

5. Multivariate Cox Proportional Hazard analysis or similar given heterogeneity in study groups e.g. (CRS/HIPEC, bimodal IV chemo, etc.)

Heterogeneity is indeed one limitation of this study as explained in detail in the discussion section.

6. For similar pathology and PCI score with differing treatments was is the objective discussion of your results?

We are not sure to correctly understand the question, our apologies. Interpretation of treatment response in patients with bi-directional treatment (if this was the question?) is indeed problematic and no definitive conclusions can be drawn.

7. PMP is usually of appendiceal carcinoma of mucinous origin (in fact you reference this in your paper).Include a comment on bromlein/NAC.

Most PMPs derive from LAMN or HAMN which are different entities. Non-operable PMPs are indeed amenable for this promising treatment alternative. This does not apply however to peritoneal metastases of appendiceal cancer = study cohort of this present analysis. We clarified this point under methods.

Round 3

Reviewer 2 Report

Dear authors - 

A few more edits:

1) The simple summary should be different than the abstract. Simplify more e.g. compared to traditional palliative chemo our median overall survival was x months vs. y months. Based on this results PIPAC for stage for peritoneal non-mucinous neoplasm is unclear/requires further study/etc.

2) Requires moderate grammar changes due to english language issues. Some sentences are incorrectly phrased.

3) Clarification point on "Bimodal IV chemo + PIPAC". There is a significant difference in the groups w/ 3 cycles of PIPAC to the other group with <3 cycles. You should comment on whether you think the improvement was due to further IV chemotherapy vs. PIPAC in the > 3 treatment vs. < 3 treatment group.

Thank you for your hard work on this paper.

Warm regards,

Author Response

1) The simple summary should be different than the abstract. Simplify more e.g. compared to traditional palliative chemo our median overall survival was x months vs. y months. Based on this results PIPAC for stage for peritoneal non-mucinous neoplasm is unclear/requires further study/etc.

The simple summary was adapted as requested.

2) Requires moderate grammar changes due to english language issues. Some sentences are incorrectly phrased.

We apologize for this and had the entire manuscript be read by an English proofreader.

3) Clarification point on "Bimodal IV chemo + PIPAC". There is a significant difference in the groups w/ 3 cycles of PIPAC to the other group with <3 cycles. You should comment on whether you think the improvement was due to further IV chemotherapy vs. PIPAC in the > 3 treatment vs. < 3 treatment group.

We thank you for this important remark. In line with our results, the use of bimodal treatment, combining PIPAC with systemic chemotherapy might be a good option to restrain tumoral progression. However, no clear data is available for appendicular cancer as of today to clearly attribute oncological efficacy to either of the two treatments. Further evidence arising from large registry data is expected in the future. We added this point on discussion session (p. 11)